# Estimating Mechanical Properties of Wood in Existing Structures—Selected Aspects

**DOI:** 10.3390/ma14081941

**Published:** 2021-04-13

**Authors:** Tomasz Nowak, Filip Patalas, Anna Karolak

**Affiliations:** Faculty of Civil Engineering, Wroclaw University of Science and Technology, Wybrzeze Wyspianskiego 27, 50-370 Wroclaw, Poland; filippatalas@wp.pl (F.P.); anna.karolak@pwr.edu.pl (A.K.)

**Keywords:** timber structures, estimating mechanical parameters, small clear specimens, non-destructive tests, semi-destructive tests, resistance drilling, ultrasonic wave, stress wave, visual grading

## Abstract

The paper presents and discusses selected methods of wood classification and the evaluation of its mechanical properties. Attention was mainly paid to methods that may be particularly useful for examining existing elements and structures. The possibility of estimating the modulus of rupture—MOR and modulus of elasticity—MOE based on the non- destructive (NDT), semi-destructive (SDT), and destructive tests (DT) were considered. Known international, European, and American standards and research approaches were indicated. The selected testing methods and their interpretation were presented. These were, among others, the method of visual assessment, the resistance drilling method, methods of determining the dynamic modulus of elasticity, and procedures for testing small clear specimens. Moreover, some of our own research results from the conducted experimental tests were presented and discussed. In the destructive tests, both large elements and small clear specimens were examined. The results obtained from individual methods were compared and some conclusions were presented. The summary discusses the fundamental difficulties and limitations in applying the presented procedures and interpretations.

## 1. Introduction

Wood is one of the oldest building materials in the world. Its widespread availability and good mechanical parameters have contributed to its wide application in civil engineering. The continued popularity of timber structures is also due to the growing interest in the use of organic materials in architecture [1].

Wood is a natural, nonhomogeneous, and anisotropic material of complex structure. Formulating a constitutive model of wood is very difficult [2]. Its mechanical parameters are influenced by many factors, among others, wood species and latewood to earlywood ratio. Moreover, in structural elements made of construction timber, the strength of the material is limited by many additional factors, such as knots (size and position), slope of grain, cracks, element size, and moisture content [3,4,5]. The precise determination of mechanical material parameters, especially in existing constructions, is a significant issue from the point of view of structural analysis. However, it is not always easy and leaves a wide freedom of interpretation. Moreover, in the case of existing structures in use, it is usually not possible to obtain much material for testing. When rebuilding, strengthening, and repairing existing structures, designers very often have the problem of assuming the proper properties and appropriate class of wood. Opposite to concrete and steel structures, where the methods of material testing are well recognized, in timber structures this problem is not clearly explained.

Researchers often use non-destructive testing (NDT) (i.e., [2,6,7,8,9,10]) or semi-destructive testing (SDT) (i.e., [11,12,13,14]). These methods do not affect the properties of the tested samples. They allow for the estimation of wood parameters without reducing the value of the tested element. In addition, a great advantage is the mobility of the used research equipment, allowing for in-situ tests when it is not possible to collect a material sample for research in a laboratory, which may be a common problem when existing and historic objects are considered [14]. Non-destructive methods also enable the detection of internal damage or material defects that may be difficult to detect with, for example, visual assessment [15]. To obtain detailed data on the physical and mechanical parameters of wood, the best method would be the use of non-destructive and destructive testing [2]. Combining the results from both methods can provide a comprehensive range of data useful for the further analysis of structural elements or entire building structures.

The aim of this article is to present selected methods of wood strength classification, which are particularly suitable for the evaluation of material in existing and historic structures.

## 2. Selected Methods for Estimation Wood Structural Properties

### 2.1. Selected Standard Procedures and Tests

There are many standards describing the procedures for in situ testing of existing and historic timber structures, including international ISO 13822 [16], European PN-EN 17121 [17], Italian UNI 11119 [18], UNI 11138 [19], and Swiss SIA 269/5 [20]. Publications of the International Council on Monuments and Sites (ICOMOS) are also widely recognized. Usually, the above standards describe the use of, non-destructive and semi-destructive methods to assess wood [14,21]. The most commonly used testing methods of NDT and SDT are presented in the diagram in Figure 1.

The aim of the research on existing structures is to obtain the most extensive and comprehensive understanding of the material structure. The applied methods give selective results and a reliable inference about the mechanical properties of the tested wood, which is possible only when many methods are combined. The literature [9,10,11,22] presents numerous examples of wood testing using the NDT and SDT methods. In addition to commonly known methods, new ones, such as air-coupled ultrasound, are also being developed [23]. The authors most often search for the correlation of NDT test results with the results of destructive tests for strength parameters. Unfortunately, publications do not always provide clear results. In the research based on acoustic methods, the correlation between MOE_dyn_ and the physical and mechanical properties of wood is sought. The analyzes (i.e., in [24,25]) most often present a strong correlation (R^2^ ≈ 0.9) between the dynamic modulus of elasticity (MOE_dyn_) obtained from acoustic methods (NDT) and the static modulus of elasticity (MOE_stat_) from destructive tests (DT). Some examples of resistance drilling tests are presented in [26]. Among the cited studies, a correlation was found between the resistance measure (RM) obtained from the resistance drilling device and density, the modulus of elasticity parallel and perpendicular to the grain, and compressive strength. The coefficients of determination R^2^ were within the range: RM-density R^2^ = 0.004–0.88, RM-MOE parallel to grain R^2^ = 0.14–0.60, RM-MOE perpendicular to grain R^2^ = 0.01–0.61, RM-compressive strength parallel to grain R^2^ = 0.52–0.64, RM-compressive strength perpendicular to grain R^2^ = 0.05–0.78. It is usually possible to collect a small amount of material from the existing structures, which can be sufficient to perform the test on small specimens as well [27]. Apart from the currently recommended tests performed on full-size elements in accordance with the applicable European standards [28,29], the literature presents numerous MOR and MOE determinations conducted on small clear specimens in accordance with the national and international standards [30,31,32,33,34,35,36]. Attempts to determine the dependence of the influence of the size of the specimens used in the research on the obtained parameter values are also made (i.e., [37,38,39,40,41,42,43]). The analyses concern the wood of trees of various exotic species (e.g., [42,43]), but also species commonly used in construction objects on the European continent, such as spruce (*Picea* sp.), pine (*Pinus* sp.) and fir (*Abies* sp.) [39]. However, the difference between the tests conducted on small specimens and the tests on structural timber should be emphasized. In the first case, the parameters of the idealized material, and in the second case, the actual building material in elements on a technical scale, are determined. Among the numerous factors affecting the mechanical parameters of wood, such factors as: different species, age of the tree from which the wood was obtained, tree growth rate, density, and local imperfections or singularities, such as cracks, knots, slope of grain, fiber deviations, depth, length etc. can be listed. The indicated material imperfections must be considered when determining the mechanical properties of wood on a technical scale on the basis of small clear specimens.

In the further part of this paper, the authors present methods of estimating the strength properties of wood based on selected methods of NDT, SDT, DT, and the method of testing small clear specimens and establishing structural properties in accordance with the American standard ASTM D245 [44].

### 2.2. Mechanical Properties Assessment Based on Visual Grading

According to the standard PN-EN 17121 [17] it is recommended that the visual assessment of wood should be based on the identification of factors that reduce its strength indicated in the standard EN-14081-1 Annex A [45]. The characteristics indicated in the standard [17], that reduce strength and can be examined in detail on site, in a non-destructive way, are knots, fiber deviation, and shrinkage gaps. It is recommended that the influence of gaps and knots be carefully assessed considering the type of structural element. Due to the great variety of rules of visual grading used in different countries, the standard [45] does not indicate a clear set of acceptable rules, only the basic criteria. Detailed descriptions of the measurement methods, classification criteria, and strength classes used should be defined at the national level. Wood can be assigned to a specific class only when all growth characteristics and properties that reduce strength are within the limits required by the class. The visual grading should be performed by qualified and experienced specialists in the field of timber structures. The rules of the visual assessment procedure may be adjusted by a specialist, provided that they are indicated in the report. According to the standard [17], the classification of existing structural elements into strength classes based on EN 338 [46] probably results in a conservative assessment. The standard [17] provides general principles for assessing existing elements. The quality class of visually graded timber is determined based on the grain, density, and species, dimensions and degree of severity of wood defects that can be seen with the unaided eye. These factors determine the strength properties of structural timber. The quality of the piece of structural timber is determined at the point of the maximum intensity of the wood defects. Depending on the quality of the wood and the quality of wood processing, according to the standard PN-D-94021 [47] the structural timber in Poland is divided into the following quality classes: KW–choice class, KS–medium quality class, KG–lower quality class. The classification is presented in Table 1.

**Table 1 materials-14-01941-t001:** The criteria for the visual grading of wood according to the PN-D-94021 [47,48].

The Classification Basis		KW (Choice Class)	KS (Middle Quality Class)		KG (Lower Quality Class)	
			Variant 1	Variant 2	Variant 1	Variant 2
Knots, regardless of quality, expressed as a knotting index USM		≤1/4	≤1/4	≤1/2	≤1/2	>1/2
Over the entire cross-section of timber USC		≤1/4	≤1/3	≤1/4	≤1/2	≤1/3
Slope of grain (diagonal grain path)		≤7% (1:14)	≤10% (1:10)		≤16% (1:6)	
Cracks, resin pockets, bark pockets and catfaces	Deep, not crossing to the face, sides and opposite plane (not including defects less than 300 mm in length)	Permissible, length up to ¼ of the piece length and not longer than 600 mm	Permissible, length up to ¼ of the piece length and not longer than 600 mm		Permissible, length up to ¼ of the piece length and not longer than 900 mm	
	Frontal non-crossing, crossing and circular	Depth up to 1/3 of the piece thickness	Depth up to 1/2 of the piece thickness		Depth up to 2/3 of the piece thickness	
Decay		Impermissible	Impermissible		Impermissible	
Insect damage		Impermissible	Impermissible		Impermissible	
Sapstain		Permissible	Permissible		Permissible	
Reaction wood (compression wood)		Permissible up to 1/5 of the girth	Permissible up to 2/5 of the girth		Permissible up to 3/5 of the girth	
Growth ring index		≤4 mm	≤6 mm		≤10 mm	
Minimum density of timber at a moisture content of 20%		≥450 kg/m^3^	≥420 kg/m^3^		≥400 kg/m^3^	
Wane is permitted along the entire length of two edges of one plane or on side, occupying a total of		Up to ¼ thickness and ¼ width of timber piece	Up to ¼ thickness and ¼ width of timber piece		(a) at a distance of up to 300 mm from faces up to 1/3 of the thickness and 1/3 of the piece width (b) at a distance of more than 300 mm from faces up to 1/2 of the thickness and 1/3 of the piece width	
Bow-longitudinal curvature of planes		≤10 mm	≤10 mm		≤20 mm	
Spring-longitudinal curvature of the sides		≤8 mm	≤8 mm		≤12 mm	
Twist in relation up to width		≤1 mm/25 mm	≤1 mm/25 mm		≤2 mm/25 mm	
Cup-cross curvature to width		≤1 mm/25 mm	≤1 mm/25 mm		≤2 mm/25 mm	
Cracks, kerf waviness		Permissible within the thickness and width deviations specified for basic dimensions				
Parallelism of planes and sides		Planes should be parallel to each other; sides of edged timber should be perpendicular to planes; deviations from parallelism should be within the limits of acceptable thickness and width deviations specified for the basic dimensions				
Non-perpendicularity of faces		Faces should be perpendicular to planes and sides; deviations from perpendicularity should be within the permissible deviations in timber length				

The classification of wood strength class can be done on the basis of PN EN 1995-1-1-NA.8.5 (Polish National Annex) [49] (Table 2). The strength class is determined directly by the relationship between the sorting class defined according to PN-D-94021 [47] and the strength class according to PN-EN 338 [46].

**Table 2 materials-14-01941-t002:** The relationship between grading classes (PN-D-94021) [47] of domestic structural timber and grading strength classes C (PN-EN 338) [46].

Tree Species	Thickness	KW (Choice Class)	KS (Middle Quality Class)	KG (Lower Quality Class)
Scots pine (*Pinus sylvestris*)	≥22 mm	C35	C24	C20
European spruce (*Picea abies*)		C30	C24	C18
European silver fir (*Abies alba*)		C22	C18	C14
European larch (*Larix decidua*)		C35	C30	C24

An alternative methodology for visual assessment is presented in the American standard ASTM D245 [44]. This standard refers directly to the results of testing small clear specimens. The influence of individual factors reducing the mechanical properties is clearly included as reducing coefficients. The standard [44] is discussed in more detail in Section 2.5, where the procedure for testing small clear specimens was presented.

### 2.3. Mechanical Properties Assessment Based on the Determination of the Dynamic Modulus of Elasticity

The dynamic modulus of elasticity of wood can be determined by various methods. Two of them are used most often: the beam vibration measurement method—the mechanical method used in strength sorting machines and the acoustic method—the stress wave or the ultrasonic wave velocity measurement.

The basic parameter required to determine the velocity of the wave propagation (v) is defined as follows:v = L/T(1)

Or
v = λ∙f(2)
where L is the distance (between two measuring points) covered by the wave; T is the time needed to cover this distance; λ is the length of the wave; and f is the frequency of the wave.

Knowing the wave propagation velocity (v) and the density of the wood (ρ), it is possible to determine the dynamic modulus of elasticity (MOE_dyn_), that can be related to the static modulus of elasticity (MOE_stat_) [50]. The dynamic modulus of elasticity can be calculated using the following formula:MOE_dyn_ = v^2^·ρ(3)
where v is the velocity of the acoustic wave and ρ is the density of the wood.

In the case of the beam vibration measurement method, the density of the tested structural timber is determined and then, by hitting the beam front, it is brought into free vibration. Measuring instruments record vibrations by determining their frequency. On the basis of the determined first harmonic information about the length of the element and the density of the wood, the dynamic modulus of elasticity (average for the element) is determined. The selected machines operating in accordance with this method are: Grade Master, Dynagrade, Mobile Timber Grader, and Visca [50].

In the case of the acoustic methods (stress wave or ultrasonic wave), devices such as the Fakopp Microsecond Timer (Fakopp Enterprise Bt., Agfalva, Hungary) or the Sylvatest (Swiss company CBS-CBT, Saint-Sulpice, Switzerland) are used. Testing with the Fakopp MS device (Fakopp Enterprise Bt., Agfalva, Hungary) (Figure 2b,c) requires the initiation of the wave with a single hit to the head with a hammer intended for this purpose. The device transmitting probes are placed in the sample, without the need to drill holes. The device measures the time of wave propagation between two transmitters. There is also a second way to measure the speed of the wave–with one transmitting probe (echo). The reflected signal is then recorded. This method significantly increases the scope of application of this method—also to elements where the access is only from one side.

When conducting the test with the Sylvatest Trio device (Figure 2a), the time of propagation of the ultrasonic wave between the transmitting and receiving probes, as well as the energy of this wave, are measured. This test requires drilling holes with a diameter of 5 mm and a depth of 10 mm in which the transmitting probes are placed. Due to the high sensitivity of the device, the measurement results may be influenced by other mechanical waves occurring near the test site, material moisture and internal stresses.

It is worth mentioning that acoustic methods require a large number of tests to eliminate measurement errors.

The velocity of the sound wave in a material is directly related to its internal structure. In the case of wood, it depends, inter alia, on the direction of the grain. Its value is several times higher parallel to the grain than perpendicular to it [50,51]. This phenomenon is due to the fact that the wave that propagates across the grain encounters more obstacles in the form of cell walls and it takes additional time to transit through them. Moreover, the examination of wood with the use of ultrasonic waves allows not only to determine MOE_dyn_, but also enables the detection of discontinuities in the material structure and assessment of its degradation. According to [51], for wood without significant structure defects, the speed of propagation of the sound wave parallel to the grain is 3500–5000 m/s, and perpendicular to the grain—1000–1500 m/s. Other values may indicate internal discontinuities in the material structure.

There are numerous attempts to correlate MOE_dyn_ with the physical and mechanical properties of wood presented in the literature. The correlation between MOE_dyn_ and MOE_stat_ obtained by destructive tests is particularly interesting. Based on the analyses presented in the literature, it can be concluded that there is a strong correlation between MOE_dyn_ and MOE_stat_ and the value of the dynamic modulus is usually about 5–15% higher than the value of the static modulus [52]. The formulas for converting the value of MOE_dyn_ to MOE_stat_ were proposed by Íñiguez-González [53] (Table 3).

**Table 3 materials-14-01941-t003:** Formulas for converting MOE_dyn_ [MPa] to MOE_stat_ [MPa] [53].

**Wood Species**	Vibration Method	Acoustic Method
Scots pine (Pinus sylvestris)	${\mathrm{MOE}}_{\mathrm{stat}}=0.9599\times {\mathrm{MOE}}_{\mathrm{dyn}}+407.2$	${\mathrm{MOE}}_{\mathrm{stat}}=0.7548\times {\mathrm{MOE}}_{\mathrm{dyn}}+579.5$
Radiata Pine (Pinus radiata)	${\mathrm{MOE}}_{\mathrm{stat}}=0.9599\times {\mathrm{MOE}}_{\mathrm{dyn}}+253.26$	${\mathrm{MOE}}_{\mathrm{stat}}=0.7548\times {\mathrm{MOE}}_{\mathrm{dyn}}-86.15$

### 2.4. Mechanical Properties Assessment Based on the Resistance Drilling Method

The drilling resistance test is a semi-destructive method that consists in drilling with a small diameter steel drill (1.5–3.0 mm) into a timber element and measuring the encountered resistance as a function of penetration depth. The drill bit advances and rotates at a constant speed. The drilling resistance corresponds to the torque required to maintain a constant drilling speed. Less torque is required in less dense areas. These are internal zones, such as the locations of corrosion, voids, gaps, and cracks. The results are presented in diagrams, examples of which are shown in Figure 3. The shape of the graph of the resistance drilling of a healthy material depends on the differences in the density of earlywood and latewood zones, the annual growth rings and the drilling angle. The most precise results are obtained by inserting the drill at an angle of 90 degrees to the annual rings and drilling in the radial direction [9,54]. The peaks in the graph indicate high drilling resistance and high density, while the dips correspond to low resistance and low density. Wood that has completely decayed or decomposed shows no resistance to drilling.

The drilling resistance method used in in situ tests enables the location of defects and internal discontinuities in timber elements without interfering with their properties. It also allows to assess the extent of wood destruction in the tested elements, to inspect the condition of wood covered with other materials (such as plaster, gypsum coatings, walls, formwork, decking, etc.), without the need to disassemble them. The weak areas and areas exposed to degradation are particularly important for testing, e.g., places where the wood contacts the ground or other materials, zones with visible moisture or biological degradation, as well as areas near door and window openings [55]. The drilling resistance method is also used in the analysis of the condition of wood in carpentry joints [56] and in elements made of glue laminated timber. However, attention should be paid to the differences in the resistographic diagrams for individual lamellas [9].

It is worth noticing that resistance drilling has a negligible effect on the mechanical and aesthetic properties of the tested element, because the diameter of the holes made during the test does not exceed 3 mm, which corresponds to the exit hole of the common wood pest in Europe-*Anobium punctatum*.

Numerous attempts are made to correlate the results of tests conducted with a resistance drilling device with the results of strength tests. Most often, based on the results of the relationship between relative resistance (RA) and drilling depth (H), the average value of the Resistance Measure (RM) parameter is determined and its relationship with the density, strength and modulus of elasticity is sought [57]. The value of RM can be calculated from the following formula:(4)$$\mathrm{RM}=\;\frac{\int_{0}^{H}\mathrm{RA}\;\times \;\mathrm{dh}}{H}$$
where: $\int_{0}^{H}\mathrm{RA}\cdot \mathrm{dh}$ is the area under the drilling resistance graph and H is the drilling depth.

### 2.5. Mechanical Properties Assessment Based on Small Clear Wood Specimens Tests

The determination of the structural timber’s characteristic values of mechanical properties and density by destructive testing may be performed in accordance with the applicable European standard PN-EN 384 + A1 [28]. According to the standard [28], structural full-size elements with defects that are representative for the population, should be tested. The standard PN-EN 384 [58] of 2011 allowed for the testing of MOR and MOE on small specimens in the case of hardwood species. The 2018 update of the standard [28] has narrowed the testing possibility to hardwood exotic species only. Moreover, it is recommended to use a minimum subsample of 40 pieces for testing.

In the case of testing single elements, particularly in existing or historic structures, examining options are very limited. In this paper, we consider testing small clear specimens (without defects) and adjusting their mechanical parameters based on ASTM D245 [44].

The calculation of the minimum quantity of samples to be tested can be performed using ISO 3129 [59] based on the determination of the testing objective, e.g., testing of a single piece of wood, the sampling method to be used and the assumed test accuracy index. According to the standard, the accuracy of 5% with a confidence level of 0.95 when determining basic physical and mechanical properties is recommended. The minimum number of samples n_min_ is calculated indicatively according to the formula:(5)$$n_{\min}\;=\;\frac{V^{2}t^{2}}{p^{2}}$$
where V is the percentage coefficient of variation for the property to be determined; t is the index of result authenticity (a half-length of the confidence interval in fractions of the standard deviation); p is the percentage index of test precision (the relation between the standard deviation of the arithmetic mean and the arithmetic mean). The average values of the coefficients of variation for basic wood properties that can be used in calculating the approximate minimum number of specimens to be taken for testing are presented in Table 4. The authors of this paper suggest taking the value of the coefficient from the column associated with ISO 3129 [59].

**Table 4 materials-14-01941-t004:** Mean coefficient of variation [%] values for main wood properties.

Wood Property	Coefficient of Variation V [%]		
	ISO 3129 [59]	Krzysik [60]	Wood Handb. [61]
Number of growth rings in 1 cm	37	-	-
Percentage of late wood	28	28	-
Density	10	10	10
Equilibrium moisture content	5	-	-
Coefficient of shrinkage: linear	28	28	-
Coefficient of shrinkage: volumetric	16	16	-
Ultimate compressive strength parallel to grain	13	13	18
Ultimate strength in static bending	15	15	16
Ultimate shearing strength parallel to grain	20	19	14
Modulus of elasticity in static bending	20	20	22
Proportional limit (conventional ultimate strength) in compression perpendicular to grain	20	30	28
Ultimate tensile strength parallel to grain	20	20	25
Ultimate tensile strength perpendicular to grain	20	-	-
Impact strength in bending	32	32	25
Hardness	17	17	20

The procedure for testing samples should be conducted in accordance with relevant standards–for MOR, for example: ISO 13061-3, PN-77/D-04103 (ISO 3133), BS 373, ASTM D143 [30,32,33,35,36], for MOE–ISO 13061-4, PN-63/D-04117, BS 373, ASTM D143 [31,33,34,35]. According to Krzysik [60], testing specimens with cross-sections ranging from 20 mm × 20 mm to 60 mm × 60 mm yields nearly equal MOR results. The author also notes that specifying the cross-sectional dimension within these limits seems to be arbitrary. Furthermore, with the increase of the support spacing, the bending strength increases within certain limits. The ratio of length to section height (l/h) is particularly important here. The ratio l/h of 10 to 15 is most commonly used for small specimens. The results increase slightly above the value l = 12 h and remain basically unchanged above l = 20 h. However, specimens with spacing less than l = 12 h are not recommended due to the effects of shear and distortion of the specimens at the locations of support and loading application. The static bending modulus according to the current testing standards for small specimens can be determined at the time of bending strength determination. There are different recommendations of the test method selection presented in the literature. According to Krzysik [60], a higher accuracy of measurement is possible to obtain in the 3-point bending test due to the larger deflections and therefore a smaller measurement error. According to BS 373 [35], the determination of MOE in cases requiring particular accuracy should be conducted in 4-point bending test, because the bending moment is constant along the section between the points of load application and, unlike in the case of 3-point bending test, there is no shear along this section, therefore there is no need to include it in the MOE calculations.

The adjustment of the strength properties of clear wood (without defects) to structural timber (with defects) can be performed according to the standard ASTM D245 [44]. The values obtained for small specimens without defects are modified by applicable factors depending on, among other things, moisture content or wood defects. General formulas for calculating mechanical properties are given below [52]:(6)$$F=l_{5}\times k_{t}\times k_{s}\times k_{p}\times k_{d}\times k_{g}\times k_{m}$$
(7)$$E=\bar{E}\times k_{t}\times k_{p}\times k_{d}\times k_{g}\times k_{m}$$
where F is the allowable stress; E is the modulus of elasticity; l_5_ is the lower 5% exclusion limit for strength; $\bar{E}$ is the mean modulus of elasticity; k_t_ is the load duration factor, k_s_ is the coefficient adjusting the characteristic values to the allowable values, k_p_ is the special factor, k_d_ is the strength ratio, dependent on wood defects, k_g_ is the special grading, k_m_ is the moisture-dependent coefficient.

A detailed discussion of the reducing factors and an example of their application can be found in the standard [44].

## 3. Materials and Methods

In the experimental part of the research, three technical scaled elements of Scots pine (*Pinus sylvestris*) with dimensions of 120 mm × 180 mm × 3600 mm were tested. The beams were initially evaluated in vibration testing and classified according to the requirements of the standard [46] into class C24.

Destructive tests were performed in a four-point bending test according to the standard PN EN 408 [29] (Figure 4a). The spacing between the supports was 3240 mm. The experimental testing was conducted in the Laboratory of Civil Engineering Structures at the Faculty of Civil Engineering of the Wroclaw University of Science and Technology. An electronically controlled linear hydraulic jack, the Instron 500 (Instron^®^, Norwood, MA, USA), was used. The results were registered using the MGC plus measurement system made by Hottinger Baldwin Messtechnik. The measurement equipment used in the experimental testing was calibrated to at least class 1 accuracy.

From the A-beams it was possible to collect 60 small specimens of clear wood without defects, which were divided into two groups. The first group (Group 1) of specimens with dimensions of 20 mm × 20 mm × 300 mm was the reference group. Specimens were tested in 3-point bending test according to the standards PN-77/D-04103, ISO 3133, ISO 13061-3, and PN-63/D-04117 [30,32,34,36]. The second group (Group 2) was the comparison group and contained specimens with dimensions of 20 mm × 20 mm × 400 mm, which were tested in a 4-point bending test. The scheme of the test stand corresponds to the testing conditions of full-size elements with defects according to the European standard PN-EN 408 [29], that finds its primary application in the testing of technical scale beams. Test schemes are shown in Figure 4 and Figure 5. The estimation of the test accuracy index values for MOR and MOE according to ISO 3129 [59] for 30 specimens is presented in Table 5.

The beams were tested by acoustic method (NDT) with the use of Fakopp Microsecond Timer and Sylvatest Trio. The reference test was performed with the Fakopp MS, taking 61 parallel and 8 perpendicular to the grain measurements for each tested beam. Additionally, for control purposes, for each beam, 5 measurements parallel and 8 perpendicular to the grain were made using the Sylvatest Trio device.

SDT tests were also conducted using the drilling resistance method. The studies were performed with the IML RESI PD-400S (IML, Wiesloch, Germany). For this, 40 drillings perpendicular to the grain were made for each beam. The drilling points were distributed evenly at both endings of the beams—every 150 mm along the lengthwise, every 40 mm width wise and 45 mm height wise, in such a way that the drilling paths did not intersect each other and not to weaken the central part of the beam. The grid of measurement points is presented in Figure 6. During the measurements, the values of Resistance Measure (RM) and Feed Force (FF) were determined.

The analyses were carried out with a reference moisture content equal to 12%. It is important to consider the significant influence of moisture content on wooden elements [14,62]. When testing wood with moisture contents differing from the reference moisture content (usually 12%)—these differences should be taken into account using the correction formulas indicated in the relevant standards.

Statistical analyses of the results were carried out using the Real Statistics Resource Pack software (Release 7.6.1). Copyright (2013–2021) Charles Zaiontz. www.real-statistics.com (accessed on 10 April 2021).

## 4. Results

### 4.1. Results of Destructive Testing of Technical Scale Beams and Density Determination

In the study of beams on a technical scale, carried out in accordance with PN-EN 408 [29], the values of MOR, MOE and density were determined and are presented in Table 6. The MOR values obtained for the beams differed significantly from each other (MOR^A01^ = 37.46 MPa, MOR^A02^ = 31.27 MPa, MOR^A03^ = 46.45 MPa). The MOE values obtained for the beams were similar (MOE^A01^ = 11.62 MPa, MOE^A02^ = 10.85 MPa, MOE^A03^ = 11.63 MPa). The MOE values determined in the destructive test and calculated on the basis of the standard PN-EN 408 [29] (Table 6) do not take into account the shear deformation. The procedure for assigning strength classes in PN-EN 384 [28] includes formulas to take into account the effect of shear deformation. The E_0_ modules determined in accordance with the standard [28], based on MOE, taking into account the influence of shear deformation, were E_0_^A01^ = 12.42 GPa, E_0_^A02^ = 11.42 GPa, E_0_^A03^ = 12.43 GPa. Based on the determined values of MOR and E_0_, it can be concluded that the beams A01, A02, A03 met the criteria of the classes C30, C24, C30 respectively [46].

The variety of mechanical properties allows for the assessment of the sensitivity of the methods used in terms of capturing these differences.

### 4.2. Results of Tests with Acoustic Method

The aim of the acoustic analysis was to determine the velocity of the wave emitted by the devices and then calculate the MOE_dyn_. The calculated values of MOE_dyn_ were used to estimate the MOE_stat_ according to Table 3. The results are shown in Table 7 and in Figure 7 and Figure 8. The velocities obtained for both devices were similar but slightly lower for the Sylvatest Trio. The MOE_stat_ values of the reference measure (Fakopp MS) parallel to the grain for beams A01, A02, A03 were 11.11 GPa, 10.70 GPa, and 10.29 GPa respectively, and differed from the results of destructive testing of technical scale beams by 4.4%, 1.4%, 11.6%. The MOE_stat_ values do not take into account the influence of shear deformation. The E_0_ modules determined in accordance with the standard [28], based on MOE_stat_ of the Fakopp MS test parallel to the grain, taking into account the influence of shear deformation, were E_0_^A01^ = 11.75 MPa, E_0_^A02^ = 11.22 MPa, E_0_^A03^ = 10.69 MPa. Based on the determined values of E_0_, it can be concluded that the beams A01, A02, and A03 met the criteria of the classes C27, C24, and C22, respectively [46].

The aim of the first statistical analysis was to check whether the differences in the mean values of velocity obtained from the basic measurement (Fakopp MS parallel to the grain) for individual beams were statistically significant. Based on the Shapiro–Wilk test, the normality assumption for all beams A01, A02, A03 was met (*p* = .242, *p* = .324, *p* = .244). There was heterogeneity of variances for all beams, as assessed by Levene’s test for equality of variances (*p* < .001). Due to the normality of the distribution of measurements and the heterogeneity of variance, we decided to perform Welch’s one-way analysis of variance (Welch’s ANOVA) with the Games–Howell post-hoc test. The conducted analysis showed a statistically significant difference in the mean velocity values measured for individual beams (*p* < .001). The post-hoc test showed a statistically significant difference in the mean values between groups A01-A03, A02-A03.

In the second statistical analysis, it was checked whether the differences in the mean values of MOE_dyn_ obtained for the basic measurement (Fakopp MS parallel to the grain) for individual beams were statistically significant. Based on the Shapiro–Wilk test, the normality assumption for all beams was met (*p* = .221, *p* = .195, *p* = .229). There was heterogeneity of variances for all beams, as assessed by Levene’s test for equality of variances (*p* < .001). Due to the normality of the distribution of measurements and the heterogeneity of variance, we decided to perform Welch’s one-way analysis of variance (Welch’s ANOVA) with the Games–Howell post-hoc test. The conducted analysis showed a statistically significant difference in the mean MOE_dyn_ values measured for individual beams (*p* < .001). The post-hoc test showed a statistically significant difference in the mean values between all beams.

### 4.3. Results of Test with Drilling Resistance Method

The purpose of testing the beams by measuring the drilling resistance was first to determine FF and RM values. The results are presented in Table 8 and Table 9 and in Figure 9, Figure 10 and Figure 11.

Based on the FF and RM values determined, statistical analyses were carried out to find the relationship between these values and the MOR, MOE, and density of elements on the technical scale.

The aim of the first statistical analysis was to check whether the differences in the median values of RM obtained in the measurement for individual beams were statistically significant. Based on the Shapiro–Wilk test, it was determined that not all distributions were normal for beams A01, A02, A03 (*p* = .250, *p* = .110, *p* = .017). There was heterogeneity of variance for all beams, as assessed by Levene’s test for equality of variance (*p* < .001). Due to the fact that not all distributions were normal, and variances were heterogeneous, we decided to perform Kruskal–Wallis test with the Games–Howell post-hoc test. The conducted analysis showed a statistically significant difference in the mean RM values measured for individual beams (*p* < .001). The post-hoc test showed a statistically significant difference in the median values between groups A01-A02, A02-A03.

In the second statistical analysis it was checked whether the differences in the median values of FF obtained in the basic measurement for individual beams were statistically significant. Based on the Shapiro–Wilk test, it was determined that not all distributions were normal for the beams (*p* = .22, *p* = .20, *p* = .23). There was heterogeneity of variance for all beams, as assessed by Levene’s test for equality of variance (*p* < .001). Due to the fact that not all distributions were normal, and variances were heterogeneous, we decided to perform Kruskal- Wallis test with the Games–Howell post-hoc test. The conducted analysis showed a statistically significant difference in the mean FF values measured for individual beams (*p* < .001). The post-hoc test showed a statistically significant difference in the median values between all beams.

Pearson correlation coefficients between FF and RM for the three different beams were equal 0.78, 0.68, and 0.76 for beams A01, A02, and A03, respectively. Figure 11b shows parallel trends for beams A02 and A03, indicating the same effects of RM on FF for those two beams. A higher trend effect was observed for beam A01. Linear regression models were fitted first for FF response and RM as the predictor separately for all three beams. The estimates from these models are presented in Table 10. The regression coefficients are significantly different from 0 for all three beams, indicating a significant correlation between FF and RM. For beam A01, with 1 unit increase of RM, FF increases by 0.96 units. For beam A02 with one unit increase of RM, FF increases by 0.55 and for beam A03 by 0.58. To investigate the differences in these trend effects between the beams, one regression model was fitted with FF as the response, RM and beam as the predictors, as well as the RM*beam interaction term. The results for this model are presented in Table 11. As can be observed, there is a significant value of beam A03 versus A01 as well as significant interaction terms confirming differences between the trend effect of RM on FF between beam A01 and the other two beams. The equality of effects for beams A02 and A03 were not tested in this model since beam A01 was the reference beam. However, in the model with changed reference beam group, the equality of these two effects were confirmed (results omitted).

Therefore, it can be concluded that the effect of RM on FF is the same for beams A02 and A03 and higher for beam A01.

To investigate the effect of density on FF and RM, the regression model was fitted with only density as the continuous predictor and FF or RM as the response. The results are presented in Table 12. Pearson correlation coefficient between density and RM was equal 0.74, between density and FF 0.76. The fitted linear trends are presented in Figure 11c,d. As can be seen, the effect of density is significant both for FF and RM. With 1 unit increase in density, both FF and RM increase by around 0.36 and 0.28 units, respectively. Similar models were fitted for FF and RM and MOR or MOE as predictors (Table 13 and Table 14). Pearson correlation coefficient between MOR and RM was equal 0.70, between MOR and FF 0.77. The fitted linear trends are presented in Figure 11e,f. Pearson correlation coefficient between MOE and RM was equal 0.76, between MOE and FF 0.79. The fitted linear trends are presented in Figure 11g,h. With 1 unit increase in MOR, both FF and RM increase by around 0.33 and 0.24 for FF and RM %, respectively (Table 13). On the other hand, with 1 unit increase in MOE, both FF and RM increase by around 5.7 and 4.4 for FF and RM %, respectively (Table 14).

### 4.4. Results of Tests of Small Clear Specimens and Their Adjustment to Structural Size Beams

The purpose of the small specimen test was to determine MOR and MOE values. The following Table 15 and Figure 12, Figure 13 and Figure 14 show the results of the conducted tests. The estimation of the 5% exclusion limit values can be done by several approaches, for example, according to ISO 12491:1997 [63] using classical statistics (EN 14358:2016 [64]) or Bayesian approach (PN-EN 1990 [65]). The 5% exclusion limit values indicated in Table 15 were determined according to PN-EN 1990 [65].

Statistical analyses were performed to determine whether there were significant differences between Group 1 and Group 2.

The aim of the first statistical analysis was to check whether the differences in the mean values of MOR between Group 1 and Group 2 were statistically significant. Based on the Shapiro–Wilk test, it was determined that both distributions were normal (*p* = .738, *p* = .134). There was heterogeneity of variance, as assessed by Levene’s test for equality of variance (*p* = .010). Due to the fact that both distributions were normal, and variances were heterogeneous, we decided to perform Welch’s *t*-test. The analysis showed no significant differences between the mean MOR values (*p* = .137).

In the second statistical analysis, we checked whether the differences in the mean values of MOE were statistically significant. Based on the Shapiro–Wilk test, it was determined that both distributions were normal (*p* = .166, *p* = .565). There was heterogeneity of variance, as assessed by Levene’s test for equality of variance (*p* < .001). Due to the fact that both distributions were normal, and variances were heterogeneous, we decided to perform a Welch’s *t*-test. The analysis showed statistically significant differences between the mean MOE values (*p* < .001).

Pearson correlation coefficients between MOE and MOR in both groups were equal 0.78 and 0.83 for Group 1 and Group 2, respectively. Linear regression models were fitted for MOR response and MOR as the predictor separately for both groups and the fitted linear trends are presented in Figure 14.

The estimates from these models are presented in Table 16. As can be seen, the regression coefficients are significantly different from zero for both groups, indicating a significant correlation between MOE and MOR. For Group 1 with 1 unit increase of MOE, MOR increases by 6.52 units and for Group 2 with one unit increase of MOE, MOR increases by 6.86. To investigate the differences in these trend effects between the groups, a regression model was fitted with MOR as the response, MOE and Group as the predictors, as well as the MOE * Group interaction term. The results for this model are presented in Table 17. As can be observed, there is no significance of Group, neither the significant interaction term. Therefore, there are no significant differences between the trend effect of MOE on MOR between Group 1 and Group 2.

This is depicted in Figure 14. The fitted trend lines are almost parallel.

The wood defects that occurred on individual beams and potentially determined their load-carrying capacity according to ASTM D245 [44] are presented in Figure 15.

The results of the MOR and MOE values adjustment determined on small clear specimens of technical scale beams using the factors indicated in ASTM D245 [44] are presented in Table 18.

Standard ASTM D245 [44] aims to define an allowable property based on formulas (6) and (7). In laboratory testing of single elements, it seems appropriate to compare the MOR and MOE of technical scale beams with the “Characteristic” and ”Mean” values of the MOR and “Mean” value of MOE estimated from small clear specimens (Table 18). The MOR “Characteristic” values of the beams were very similar in both groups. The MOE values estimated in Group 1 for beams A01, A02, A03 were 11.7 GPa, 10.5 GPa, 11.7 GPa respectively, and differed from the values obtained in the destructive test of technical scale beams by 0.7%, 3.2%, 0.7%. The MOE values estimated in Group 2 for beams A01, A02, A03 were 13.8 GPa, 12.4 GPa, 13.8 GPa respectively, and differed significantly from the values obtained in the destructive test of technical - the differences were 18.8%, 14.3%, 18.8% respectively. The MOE values do not consider the shear deformation. The procedure for assigning strength classes in PN-EN 384 [28] includes formulas to take into account the effect of shear deformation. The E_0_ modules determined in accordance with the standard [28], based on MOE, taking into account the influence of shear deformation, were for Group 1: E_0_^A01^ = 12.52 GPa, E_0_^A02^ = 11.00 GPa, E_0_^A03^ = 12.52 GPa, for Group 2: E_0_^A01^ = 15.25 GPa, E_0_^A02^ = 13.46 GPa, E_0_^A03^ = 15.25 GPa. Based on the determined values of MOR (“CHARACTERISTIC”) and E_0,_ it can be concluded that the beams A01, A02, A03 for Group 1 met the criteria of the classes C30, C24, C30 respectively, and for Group 2: C30, C27, C35 [46].

**Table 18 materials-14-01941-t018:** Estimation of MOR and MOE of technical scale beams from small clear specimens according to the standard ASTM [44].

	Beam A01				Beam A02				Beam A03			
	Group 1		Group 2		Group 1		Group 2		Group 1		Group 2	
	MOR	MOE	MOR	MOE	MOR	MOE	MOR	MOE	MOR	MOE	MOR	MOE
Mean [MPa]	96	11,700	103	13,800	96	11,700	103	13,800	96	11,700	103	13,800
5% exclusion value (MPa)	80	9800	66	9400	80	9800	66	9400	80	9800	66	9400
Special factor k_p_ ^1^ [-]	0.68	-	0.78	-	0.68	-	0.78	-	0.68	-	0.78	-
Factor k_d_ ^2^ [-]	0.60	1.0	0.60	1.0	0.53	0.90	0.53	0.90	0.75	1.00	0.75	1.00
5% exclusion value $5\%\;\mathrm{exclusion}\;\mathrm{value}\;\times k_{p}\times k_{d}$[MPa] “Characteristic"	33	-	31	-	29	-	27	-	41	-	39	-
$\mathrm{Mean}\;\times k_{p}\times k_{d}$[MPa] “Mean”	39	11,700	48	13,800	35	10500	43	12,400	49	11,700	60	13,800

^1^ k_p_—special factor for 4-point bending was determined based on the formula proposed in [61]. ^2^ k_d_—strength ratio dependent on wood defects [44].

## 5. Discussion

The main objective of this paper was to attempt to indicate the practical aspect of testing timber elements in existing structures, especially historic ones, and to interpret the results. A strict analysis of three elements with similar MOE but significantly different MOR values is presented. The focus was on the interpretation of the results oriented towards single elements, which also refers to the study of historic structures, where sometimes even a single element subjected to multiple tests may be of interest. The authors were mainly interested in practical conclusions, while statistical analyses were carried out to complete the research overview. The authors were concerned to indicate that NDT and SDT methods should not be used selectively (alone) in timber assessment.

Those testing methods were chosen which were considered to be the most applicable, fast, useful in-situ, and uncomplicated in terms of conducting the tests and processing the results. The acoustic method and resistance drilling are well known in the literature. Especially the reliability and effectiveness of the former has been proven [24,25]. Most often, the literature looks for correlations between the results obtained and the mechanical properties, while the practical aspect of interpreting the results is neglected. In addition, the standards for testing existing structures are not unambiguous and the interpretation of the research results is largely unsystematic.

Studies reported in the literature indicate strong correlations between MOE_dyn_ values obtained from the acoustic method and actual MOE_stat_ values from DT testing. Based on the study, MOE_stat_ values differing from the actual values within 5% were obtained for beams A01 and A02, but for beam A03 the difference was more than 11%. Such a difference in MOE_stat_ values can result in incorrect timber assignment, even by several classes.

The resistance drilling method study indicated the presence of statistically significant correlations between FF and RM values and density, MOE and MOR, but they are so weak that it seems inappropriate to infer strength properties on their basis. Nevertheless, the drilling resistance method has excellent applications in qualitative wood assessment—e.g., determining the degree of biological degradation and finding hidden defects within the element.

In addition, small samples without defects were tested and the results were interpreted in accordance with ASTM D245. The use of ASTM D245 in combination with EN and ISO standards, especially for testing historic elements, seems unique but possible. By using simple statistical methods and the methodology of reducing the strength properties of clear wood without defects based on ASTM D245, good results and an accurate classification in the reference group (Group 1) were obtained. However, it should be noted that it is not always possible to extract enough material from an existing structure for testing and accurate visual assessment is not always possible.

## 6. Conclusions

The primary goal of this paper was to consider possible methods useful in determining MOR and MOE in existing structures where testing capabilities are limited. Attention is drawn to the problem of obtaining a sufficient amount of material for testing and comprehensive assessment of wood. On the basis of the conducted non-destructive, semi-destructive and destructive tests, it was possible to determine the following conclusions:In the conducted research, very good correlations were obtained between MOE_stat_ from the acoustic method and MOE from beams on the technical scale. Nevertheless, the difference in the value of the modulus determined by the acoustic method may be high enough to result in an incorrect assignment of wood class.In the drilling resistance test, statistically significant correlations were observed between the density, MOR and MOE as predictors and FF or RM as the response (R^2^ = 0.49–0.62). However, it is considered that resistance drilling should be used for qualitative rather than quantitative evaluation of timber.In the conducted study, the MOE values determined in accordance with the ASTM standard, based on the results for small clear specimens, correspond very well with the actual values from the technical scale element tests.The failure of the technical scale beams in the bending tests was observed in the areas of occurrence of defects considered by ASTM D245 [44] to be critical for load-bearing capacity.It is recommended to use different methods in parallel, as no single method is sufficiently reliable.

It should be noted that the research was conducted on elements made of pine (*Pinus sylvestris*), which, due to its good availability, low price, and good strength properties, is the most popular structural wood in Poland [66]. The results of wood testing should be considered in the context of a specific species.

Numerous ISO, EN and ASTM standards were used in the analyses. Special care should be taken when combining the standards. The material properties to be compared should be determined from the analogous, well-known formulas and the relationships from materials mechanics.

The laboratory tests and analyses presented in this paper are part of an ongoing research project. Due to limited data, the conclusions and observations presented should be considered possible but not certain.

## Figures and Tables

**Figure 1 materials-14-01941-f001:**
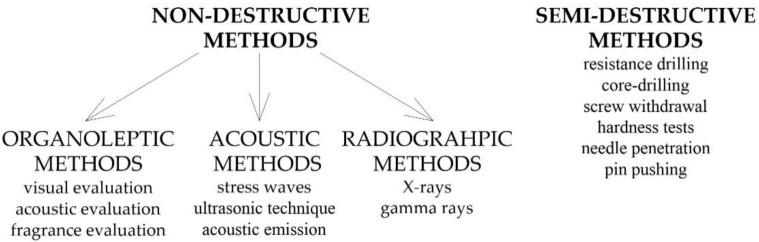
Non-destructive and semi-destructive testing methods.

**Figure 2 materials-14-01941-f002:**
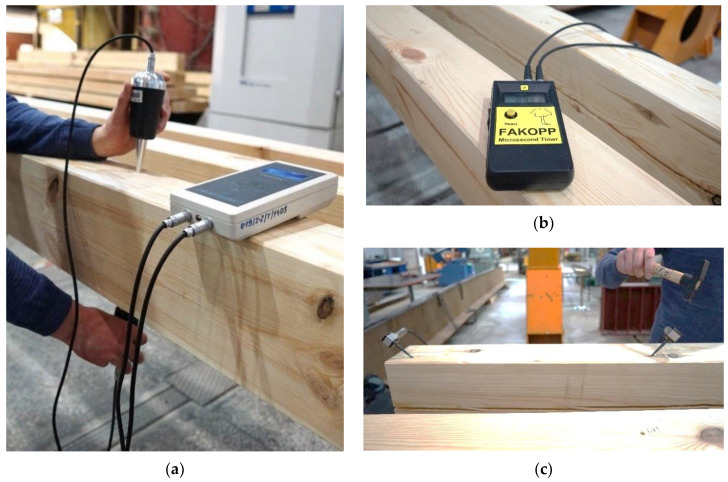
Determination of the dynamic modulus of elasticity using the acoustic method: (**a**) testing with the Sylvatest Trio device, (**b**,**c**) testing with the Fakopp MS device

**Figure 3 materials-14-01941-f003:**
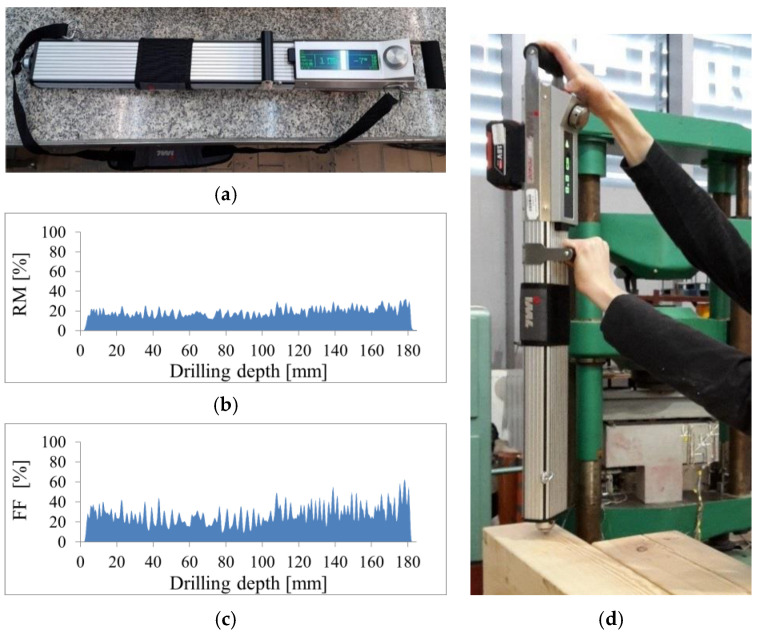
Drilling resistance test (**a**) device IML RESI PD-400S used in tests (**b**,**c**) charts obtained from test (**d**) test procedure.

**Figure 4 materials-14-01941-f004:**
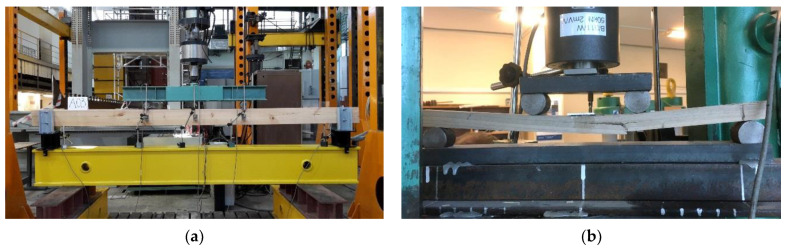
View of experimental stands for testing: (**a**) technical scale beam, (**b**) small clear specimen (Group 2).

**Figure 5 materials-14-01941-f005:**
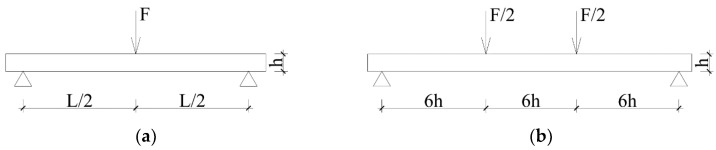
Scheme of testing: (**a**) small specimens Group 1, (**b**) small specimens Group 2 and technical scale beams, where F—load, L—span in bending, h—height of the beam.

**Figure 6 materials-14-01941-f006:**
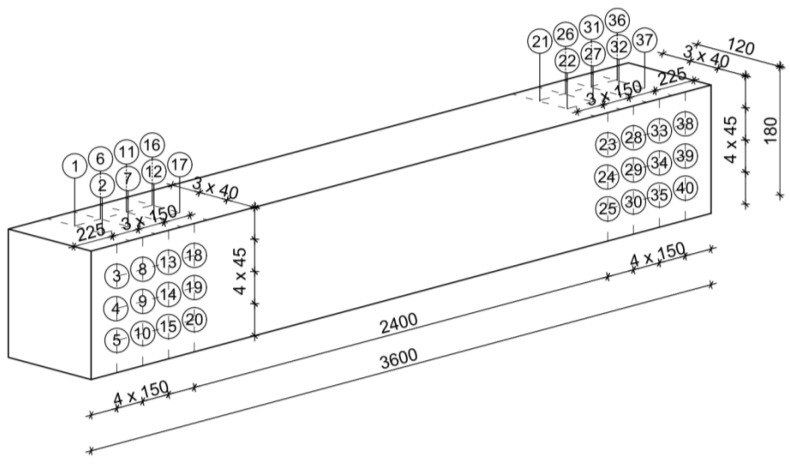
Diagram of the location of drilling points, unit: [mm].

**Figure 7 materials-14-01941-f007:**
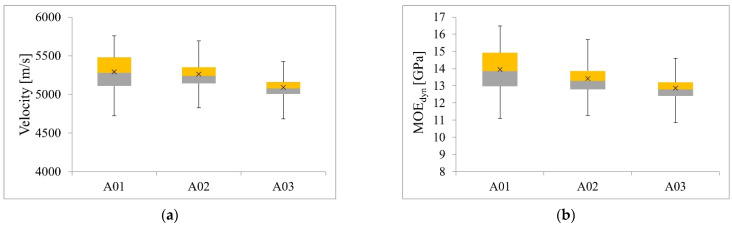
Box plots for Fakopp MS measurement parallel to the grain: (**a**) velocity, (**b**) MOE_dyn_.

**Figure 8 materials-14-01941-f008:**
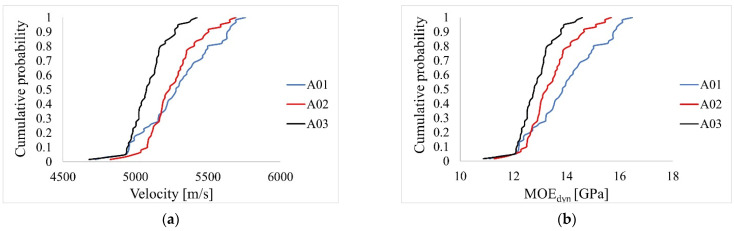
Empirical cumulative distribution function for Fakopp MS measurement parallel to the grain: (**a**) velocity, (**b**) MOE_dyn_.

**Figure 9 materials-14-01941-f009:**
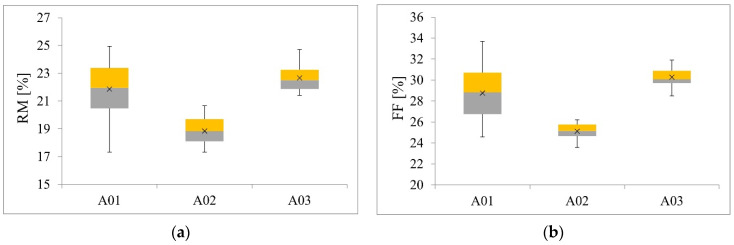
Box plots: (**a**) Resistance Measure–RM, (**b**) Feed Force–FF.

**Figure 10 materials-14-01941-f010:**
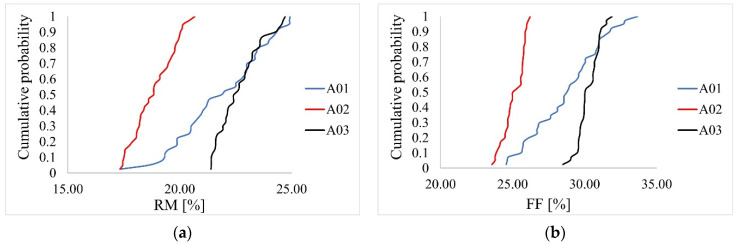
Empirical cumulative distribution function: (**a**) Resistance Measure–RM, (**b**) FF–Feed Force.

**Figure 11 materials-14-01941-f011:**
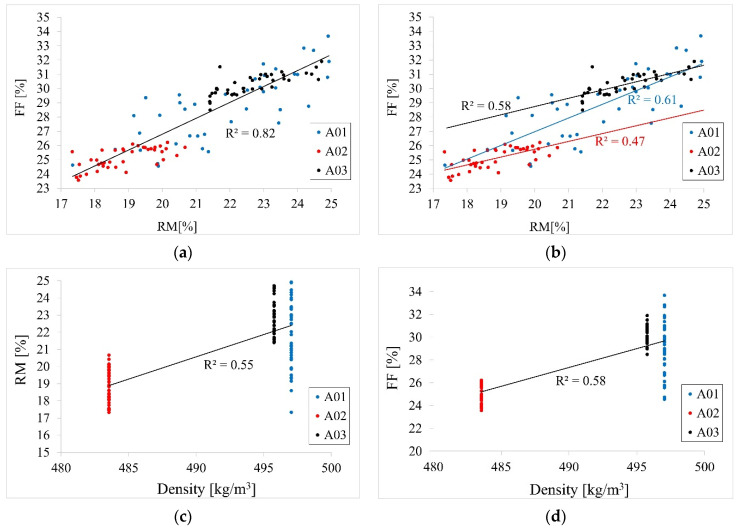
Correlation plots of Resistance Measure and Feed Force results (**a**) correlation between RM and FF, (**b**) correlation between RM and FF for beams separately, (**c**) correlation between RM and density, (**d**) correlation between FF and density, (**e**) correlation between RM and MOR obtained from structural size beam bending test, (**f**) correlation between FF and MOR obtained from structural size beam bending test, (**g**) correlation between RM and MOE obtained from structural size beam bending test, (**h**) correlation between FF and MOE obtained from structural size beam bending test.

**Figure 12 materials-14-01941-f012:**
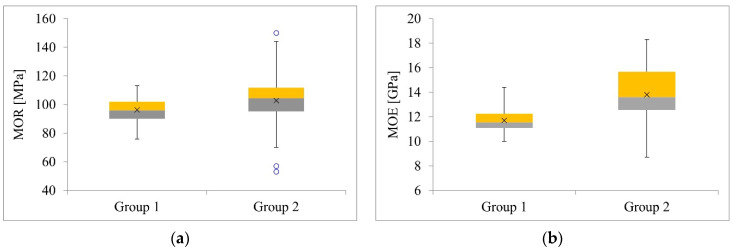
Box plots: (**a**) MOR, (**b**) MOE.

**Figure 13 materials-14-01941-f013:**
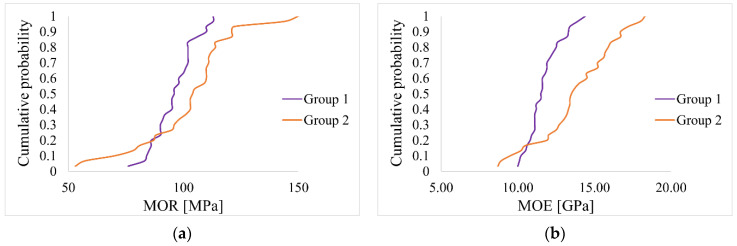
Empirical cumulative distribution function: (**a**) MOR, (**b**) MOE.

**Figure 14 materials-14-01941-f014:**
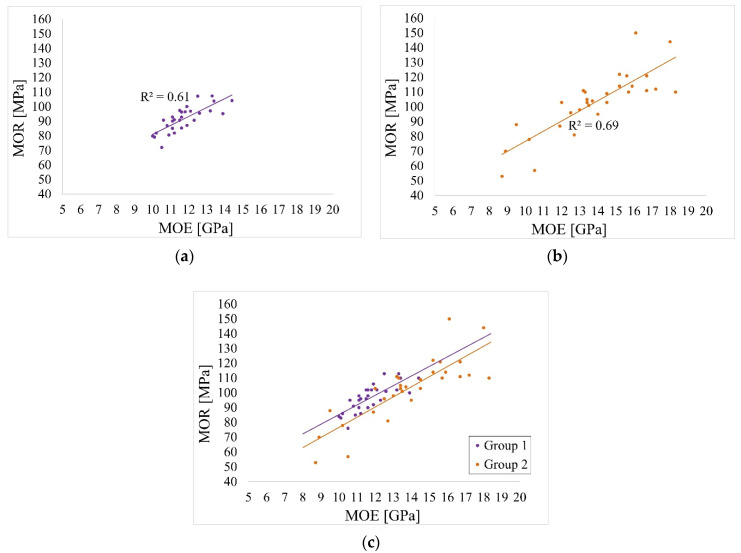
Correlation between MOE and MOR: (**a**) Group 1, (**b**) Group 2, (**c**) both Groups.

**Figure 15 materials-14-01941-f015:**
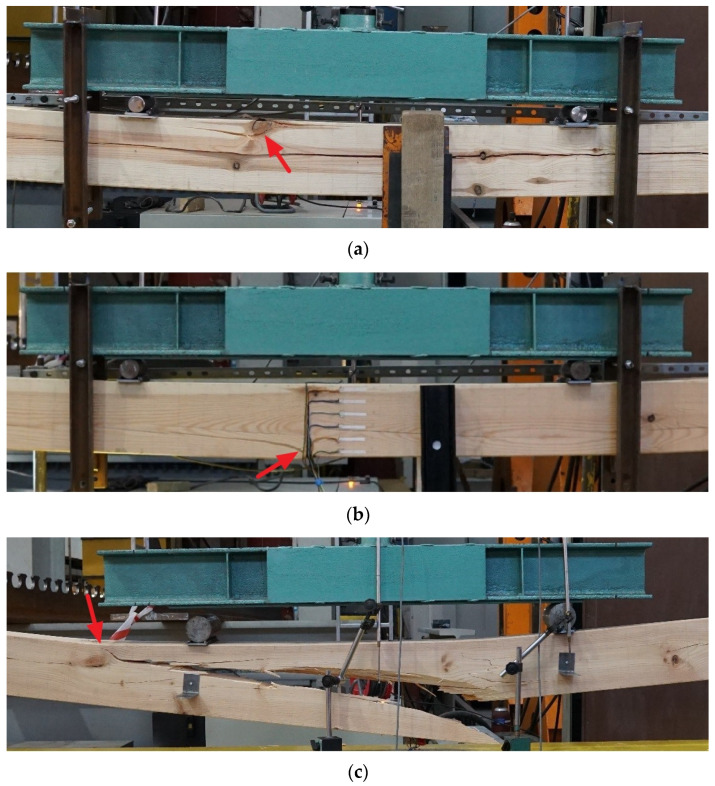
Defects indicated according to ASTM D245 [44] and potentially determining the load-bearing capacity of the beams: (**a**) beam A01-damage initiation caused by crushing of compressed fibers in the corner knot area (compression near knot [52]), (**b**) beam A02-damage initiation caused by rupture of tensioned fibers near the corner knot and slope of grain (cross grain tension/diagonal tension [33,52]) (**c**) beam A03-damage initiation caused by crushing of compressed fibers in corner knot area (cross grain tension/diagonal tension caused by compression near knot [33,52]).

**Table 5 materials-14-01941-t005:** Estimation of the test accuracy rates.

	MOR	MOE
Number of samples	30	30
Coefficient of variation according to ISO 3129	15%	20%
Confidence level	0.95	0.95
Estimated test accuracy rate	5.37%	7.16%

**Table 6 materials-14-01941-t006:** The results of bending technical scale beams and values of their density.

Beam	MOR				MOE				Density [kg/m^3^]
	Value [MPa]	Mean [MPa]	Standard Deviation [MPa]	Variation Coefficient ν [%]	Value [GPa]	Mean [GPa]	Standard Deviation [GPa]	Variation Coefficient ν [%]	
A01	37.46	38.39	6.23	16.23	11.62	11.36	0.37	3.23	497
A02	31.27				10.85				484
A03	46.45				11.63				496

**Table 7 materials-14-01941-t007:** The results of the acoustic tests with Fakopp MS and Sylvatest Trio- mean values.

Beam	Direction Relative to Grain	Velocity [m/s]		MOE_dyn_ [GPa]		MOE_stat_ ^1^ [GPa]	
		Fakkop MS	Sylvatest Trio	Fakkop MS	Sylvatest Trio	Fakkop MS	Sylvatest Trio
A01	parallel	5292	5134	13.95 ^2^Δ = 20.1%	13.10 Δ = 12.7%	11.11 Δ = 4.4%	10.47 Δ = 9.9%
	perpendicular	1689	1617	1.42	1.31	1.65	1.57
A02	parallel	5262	5155	13.41 Δ = 23.6%	12.87 Δ = 18.6%	10.70 Δ = 1.4%	10.29 Δ = 5.1%
	perpendicular	1742	1559	1.47	1.18	1.69	1.47
A03	parallel	5090	5038	12.86 Δ = 10.6%	12.59 Δ = 8.3%	10.29 Δ = 11.6%	10.08 Δ = 13.3%
	perpendicular	1668	1573	1.38	1.23	1.62	1.51

^1^ MOE_stat_ was determined by the formula in Table 3. ^2^Δ are the percentages of the difference of MOE determined from acoustic methods with MOE values obtained by destructive testing of technical scale beams (Table 6).

**Table 8 materials-14-01941-t008:** Mean drilling resistance test results.

Beam	Number of Measurements	Resistance Measure RM [%]			
		Mean	Range	Standard Deviation	Coefficient of Variation
A01	40	21.86	17.34–24.93	1.96	8.99
A02	40	18.84	17.33–20.67	0.95	5.04
A03	40	22.66	22.66–21.40	0.99	4.38
Summary	120	21.12	17.33–24.93	2.15	10.17

**Table 9 materials-14-01941-t009:** Mean feed force test results.

Beam	Number of Measurements	Feed Force FF [%]			
		Mean	Range	Standard Deviation	Coefficient of Variation
A01	40	28.76	24.56–33.68	2.41	8.39
A02	40	25.11	23.57–26.22	0.76	3.02
A03	40	30.28	28.49–31.90	0.75	2.48
Summary	120	28.05	23.57–33.68	2.65	9.44

**Table 10 materials-14-01941-t010:** Estimates for regression models of FF vs. RM for different beams.

Beam	Variable	Estimate	Standard Error	*p*-Value
A01	Intercept	7.79	2.73	.007
	RM	0.96	0.12	<.001
A02	Intercept	14.78	1.78	<.001
	RM	0.55	0.09	<.001
A03	Intercept	17.19	1.80	<.001
	RM	0.58	0.08	<.001

**Table 11 materials-14-01941-t011:** Estimates for regression models of FF with RM beam as the predictor and RM*Beam interaction term.

Variable	Estimate	Standard Error	*p*-Value
Intercept	7.79	1.76	<.001
RM	0.96	0.08	<.001
Beam A02 vs. A01	6.99	3.58	.053
Beam A03 vs. A01	9.40	4.00	.021
RM: beam A02 vs. A01	−0.41	0.18	.027
RM: beam A03 vs. A01	−0.38	0.18	.034

**Table 12 materials-14-01941-t012:** Estimates for regression models of FF and RM with density as the predictor.

Feed Force/Resistance Measure	Variable	Estimate	Standard Error	*p*-Value
FF	Intercept	−148.28	13.79	<.001
	Density	0.36	0.03	<.001
RM	Intercept	−117.11	11.68	<.001
	Density	0.28	0.02	<.001

**Table 13 materials-14-01941-t013:** Estimates for regression models of FF and RM with MOR as the predictor.

Feed Force/Resistance Measure	Variable	Estimate	Standard Error	*p*-Value
FF	Intercept	15.47	0.96	<.001
	MOR	0.33	0.02	<.001
RM	Intercept	11.90	0.88	<.001
	MOR	0.24	0.02	<.001

**Table 14 materials-14-01941-t014:** Estimates for regression models of FF and RM with MOE as the predictor.

Feed Force/Resistance Measure	Variable	Estimate	Standard Error	*p*-Value
FF	Intercept	−36.78	4.64	<.001
	MOE	5.70	0.41	<.001
RM	Intercept	−29.19	4.01	<.001
	MOE	4.43	0.35	<.001

**Table 15 materials-14-01941-t015:** Results of small clear specimen bending tests.

	Group 1		Group 2	
	MOR	MOE	MOR	MOE
Number of specimens	30 pieces		30 pieces	
Mean value	96 MPa	11.7 GPa	103 MPa	13.8 GPa
Standard deviation	9.3 MPa	1.11 GPa	21.2 MPa	2.57 GPa
Coefficient of variation	9.6%	9.5%	20.2%	18.3%
Confidence interval for the mean (0.95)	92.7–99.3 MPa	11.43–12.23 GPa	95.1–110.3 MPa	12.88–14.72 GPa
5% exclusion limit	80 MPa	9.8 GPa	66 MPa	9.4 GPa

**Table 16 materials-14-01941-t016:** Estimates for regression models of MOR vs. MOE for Groups 1 and 2.

Group	Variable	Estimate	Standard Error	*p*-Value
Group 1	Intercept	19.99	11.52	0.094
	MOE	6.52	0.98	<0.001
Group 2	Intercept	8.08	12.07	0.509
	MOE	6.86	0.86	<0.001

**Table 17 materials-14-01941-t017:** Estimates for regression models of MOR with MOE, Group as the predictor and MOE * Group interaction term.

Variable	Estimate	Standard Error	*p*-Value
Intercept	19.99	18.44	0.283
MOE	6.52	2.07	<0.001
Group 2 vs. 1	−11.91	20.75	0.568
MOE * Group	0.33	1.71	0.846

## Data Availability

Data sharing not applicable.

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
