# Peer review of "Estimating Mechanical Properties of Wood in Existing Structures—Selected Aspects"

_materials, 2021, doi:10.3390/ma14081941_

Round 1

Reviewer 1 Report

This manuscript deals with Estimating Mechanical Properties of Wood in Existing Structures - Selected Aspects.

This is an interesting topic and research.
The manuscript builds on the authors' previous research.

The submitted manuscript is a new version of the previous (rejected) manuscript.

The authors have significantly improved the manuscript and the experimental program has sufficient scope.

The topic itself is solved logically.

However, the manuscript can be further improved.

The introduction part should be extended. Currently, extensive research is devoted to wood and wooden structures. The addressed topic must be more placed in the context of the current state. It would be appropriate to expand also more references eg. about experiment testing with diagnostic or testing of woods elements. 

Vavrusova, K. et. al. Analysis of Longitudinal Timber Beam Joints Loaded with Simple Bending. Sustainability 2020, 12, 9288.
Oliveira, J. et. al.  Direct Evaluation of Mixed Mode I+II Cohesive Laws of Wood by Coupling MMB Test with DIC. Materials 2021, 14, 374. 

Part of the discussion should be mentioned separately. It is necessary to better present new knowledge from research.

New findings and benefits of research on why to read the article must be clearly stated.

After editing the manuscript, it will present a potentially interesting article for readers of the Materials journal.

The manuscript must be revised before publication.

Author Response

The authors would like to thank the Reviewer for agreeing to review our manuscript and for their very helpful comments and suggestions. The paper was corrected according to the revision. We have responded to all the suggestions and recommendations of the Reviewer. Detailed responses to the specific points are listed below.

Comments and Suggestions for Authors

This manuscript deals with Estimating Mechanical Properties of Wood in Existing Structures - Selected Aspects.

This is an interesting topic and research.

The manuscript builds on the authors' previous research.

The submitted manuscript is a new version of the previous (rejected) manuscript.

The authors have significantly improved the manuscript and the experimental program has sufficient scope.

The topic itself is solved logically.

However, the manuscript can be further improved.

The introduction part should be extended. Currently, extensive research is devoted to wood and wooden structures. The addressed topic must be more placed in the context of the current state. It would be appropriate to expand also more references eg. about experiment testing with diagnostic or testing of woods elements. 

Vavrusova, K. et. al. Analysis of Longitudinal Timber Beam Joints Loaded with Simple Bending. Sustainability 2020, 12, 9288.
Oliveira, J. et. al.  Direct Evaluation of Mixed Mode I+II Cohesive Laws of Wood by Coupling MMB Test with DIC. Materials 2021, 14, 374. 

Introduction has been modified according to the recommendations (lines 29-30, 34-36):

“The continued popularity of timber structures is also due to the growing interest in the use of organic materials in architecture [1].”

“Moreover, in structural elements made of construction timber, the strength of the material is limited by many additional factors, such as knots (size and position), slope of grain, cracks, element size, moisture content [2,3,4].”

Added references:

  1. Šubic, B.; Fajdiga, G.; Lopatič, J. Bending Stiffness, Load-Bearing Capacity and Flexural Rigidity of Slender Hybrid Wood-Based Beams. Forests 2018, 9, 703. https://doi.org/10.3390/f9110703
  2. Baño, V.; Arriaga, F.; Guaita, M. Determination of the influence of size and position of knots on load capacity and stress distribution in timber beams of Pinus sylvestris using finite element model. Biosyst. Eng. 2013, 114 (3), 214-222, https://doi.org/10.1016/j.biosystemseng.2012.12.010.
  3. Vavrusova, K.; Lokaj, A.; Mikolasek, D.; Sucharda, O. Analysis of Longitudinal Timber Beam Joints Loaded with Simple Bending. Sustainability 2020, 12, 9288. https://doi.org/10.3390/su12219288
  4. Oliveira, J.; Xavier, J.; Pereira, F.; Morais, J.; de Moura, M. Direct Evaluation of Mixed Mode I+II Cohesive Laws of Wood by Coupling MMB Test with DIC. Materials 2021, 14, 374. https://doi.org/10.3390/ma14020374

Part of the discussion should be mentioned separately. It is necessary to better present new knowledge from research.

Discussion has been separated in the new section 5 as recommended by the Reviewer.

New findings and benefits of research on why to read the article must be clearly stated.

The benefits of the article are indicated in the section 5.

After editing the manuscript, it will present a potentially interesting article for readers of the Materials journal.

The manuscript must be revised before publication.

Reviewer 2 Report

In general terms, we consider that the authors have replied satisfactorily to the comments and suggestions provided by this reviewer. Several doubtful questions have been clarified, providing additional information on the development of the tests or on the estimation of characteristic values. In this sense, the content of the new section 4.5 is especially valued. All of this has contributed to improving the scientific quality of the document. However, some cautions can be further stated.

We agree to partially modify the title. However, we doubt that "selected aspects" adequately illustrates the main objectives of the research.

In response to our considerations on the reduced number of pieces tested, the authors have indicated: “The main idea was to consider the practical aspect of the applicability of the presented methods”. Despite this, the article assesses possible correlations between the results obtained by different techniques with the mechanical properties of wood. Consequently, several statistical strategies are used. And that is why we must insist on the need to use larger population samples to increase statistical significance in the analysis and in the formulation of conclusions. At the end of the article it is emphasized that the above is part of an ongoing research project: “Due to limited data, the conclusions and observations presented should be considered possible but not certain” (line 658). This statement seems consistent with our opinion, and for this reason we hope that the authors will be able to contribute new data in the near future, overcoming these limitations.

We also maintain our point of view in relation to the drafting of conclusions. We refer to the substitution of purely qualitative assessments ("correspond very well”, lines 635 and 642) by quantitative ones, based on the observed results. On the other hand, the first conclusion (line 635) is contradictory in relation to what is indicated in the new section 4.5, where it is stated that the MOEstat values may result in incorrect timber assignment.

Author Response

The authors would like to thank the Reviewer for agreeing to review our manuscript and for their very helpful comments and suggestions. The paper was corrected according to the revision. We have responded to all the suggestions and recommendations of the Reviewer. Conclusions were also corrected and extended. Detailed responses to the specific points are listed below.

Comments and Suggestions for Authors

In general terms, we consider that the authors have replied satisfactorily to the comments and suggestions provided by this reviewer. Several doubtful questions have been clarified, providing additional information on the development of the tests or on the estimation of characteristic values. In this sense, the content of the new section 4.5 is especially valued. All of this has contributed to improving the scientific quality of the document. However, some cautions can be further stated.

We agree to partially modify the title. However, we doubt that "selected aspects" adequately illustrates the main objectives of the research.

In response to our considerations on the reduced number of pieces tested, the authors have indicated: “The main idea was to consider the practical aspect of the applicability of the presented methods”. Despite this, the article assesses possible correlations between the results obtained by different techniques with the mechanical properties of wood. Consequently, several statistical strategies are used. And that is why we must insist on the need to use larger population samples to increase statistical significance in the analysis and in the formulation of conclusions. At the end of the article it is emphasized that the above is part of an ongoing research project: “Due to limited data, the conclusions and observations presented should be considered possible but not certain” (line 658). This statement seems consistent with our opinion, and for this reason we hope that the authors will be able to contribute new data in the near future, overcoming these limitations.

We also maintain our point of view in relation to the drafting of conclusions. We refer to the substitution of purely qualitative assessments ("correspond very well”, lines 635 and 642) by quantitative ones, based on the observed results. On the other hand, the first conclusion (line 635) is contradictory in relation to what is indicated in the new section 4.5, where it is stated that the MOEstat values may result in incorrect timber assignment.

The text has been modified in accordance with the recommendation (lines 637-640, 645-647):

In the conducted research, very good correlations were obtained between MOEstat from the acoustic method and MOE from beams on the technical scale. Nevertheless, the difference in the value of the modulus determined by the acoustic method may be high enough to result in an incorrect assignment of wood class.”

“In the conducted study, the MOE values determined in accordance with the ASTM standard, based on the results for small clear specimens, correspond very well with the actual values from the technical scale element tests.”
